# Root-Shoot Nutrient Dynamics of Huanglongbing-Affected Grapefruit Trees

**DOI:** 10.3390/plants11233226

**Published:** 2022-11-24

**Authors:** Lukas M. Hallman, Davie M. Kadyampakeni, John-Paul Fox, Alan L. Wright, Lorenzo Rossi

**Affiliations:** 1Indian River Research and Education Center, Horticultural Sciences Department, Institute of Food and Agricultural Sciences, University of Florida, Fort Pierce, FL 34945, USA; 2Citrus Research and Education Center, Soil, Water and Ecosystem Sciences Department, Institute of Food and Agricultural Sciences, University of Florida, Lake Alfred, FL 33850, USA; 3Indian River Research and Education Center, Soil, Water and Ecosystem Sciences Department, Institute of Food and Agricultural Sciences, University of Florida, Fort Pierce, FL 34945, USA

**Keywords:** citrus greening, *Citrus paradisi*, destructive analysis, nutrient translocation, fertilizer application methods

## Abstract

With huanglongbing (HLB) causing a reduction in fine root mass early in disease progression, HLB-affected trees have lower nutrient uptake capability. Questions regarding the uptake efficiency of certain fertilizer application methods have been raised. Therefore, the goals of this study are to determine if nutrient management methods impact nutrient translocation and identify where in the tree nutrients are translocated. Destructive nutrient and biomass analysis were conducted on field grown HLB-affected grapefruit trees (*Citrus × paradisi*) grafted on ‘sour orange’ (*Citrus × aurantium*) rootstock under different fertilizer application methods. Fertilizer was applied in the form of either 100% soluble granular fertilizer, controlled release fertilizer (CRF), or liquid fertilizer. After three years, the entire tree was removed from the grove, dissected into eight different components (feeder roots, lateral roots, structural roots, trunk, primary branches, secondary branches, twigs, and leaves), weighed, and then analyzed for nutrient contents. Overall, application methods showed differences in nutrient allocation in leaf, twig, and feeder root; however, no consistent pattern was observed. Additionally, leaf, twig, and feeder roots had higher amount of nutrients compared to the other tree components. This study showed that fertilization methods do impact nutrient contents in different components of HLB-affected trees. Further research should be conducted on the impact of different fertilizer application methods and rates on HLB-affected trees.

## 1. Introduction

Proper nutrient management is integral to increase both the operational profitability and environmental sustainability of Florida’s citrus industry. Citrus trees require 17 elements for optimum growth and production [1]; of which, nitrogen (N), phosphorus (P), potassium (K), magnesium (Mg), calcium (Ca), sulfur (S), iron (Fe), zinc (Zn), manganese (Mn), copper (Cu), and boron (B) are commonly supplied via inorganic fertilizers. If any essential element is deficient, tree growth, development, and yield is reduced [2,3,4]. Although fertilization is commonplace, supplying the optimum levels of nutrients can be challenging due to several factors such as edaphic conditions, environmental issues, and disease pressure [5,6].

Oftentimes sandy soils in citrus growing regions, such as those found in Florida, are low in natural fertility, resulting in required higher and more frequent fertilizer inputs [7,8,9]. This management method is not without risk since the over application of fertilizers can result in toxicity effects on the plants and nutrient leaching into groundwater [10,11].

Huanglongbing (HLB; Citrus greening) impacts nutrient uptake and translocation. This disease is caused by the phloem limited bacteria *Candidatus* Liberibacter asiaticus (*C*Las) and vectored by the Asian citrus Psyllid *Diaphorina citri*; moreover, it is commonly associated with root and canopy dieback, poor fruit yield and quality, and tree death [12,13]. Research has also shown that HLB alters nutrient contents within affected trees [14,15]. For example, a study by da Silva et al. [16] found that HLB-affected sweet orange (*Citrus × aurantium*) trees had lower levels of N, Mg, and S in leaves and sap extracts. Similarly, a study by Shahzad et al. [17] found that HLB-affected sweet orange trees had lower leaf Ca, Mg, and S compared to healthy trees.

The studies described above clearly demonstrate that HLB changes how nutrients are taken up by and translocated within affected trees. Although much research has been focused on how different nutrient rates impact HLB-affected tree health [18,19,20,21], there is a lack of understanding regarding where in the trees these nutrients are translocated. For example, a previous study conducted by [22] analyzed nutrient contents on different components of sweet orange trees, but this was done with non HLB-affected trees.

In addition to applying the correct rate of fertilizers, the method in which fertilizers are applied can be equally important. In the Florida citrus industry, both liquid and solid granular fertilizers are readily available and commonly used [23]. Granular fertilizers can be formulated as 100% soluble or controlled release fertilizers (CRF). The nutrients in CRF have polymer coatings which better control nutrient release timing and rate compared to 100% soluble granular fertilizers [24,25]. Liquid fertilization is also popular since it can be integrated into existing irrigation systems. This integration allows for more precise control over fertilizer rate, timing, and location. The use of liquid fertilization has been shown to reduce nutrient leaching, increase tree growth rates, and increase yield [23,26]. Very few studies exist that compare the use of 100% soluble granular fertilizers, CRF, and liquid fertilization on HLB-affected grapefruits (*Citrus* × *paradisi*) grown on flatwood soils; furthermore, none of the existing studies conducted a full tree destructive biomass and nutrient content analysis. 

As HLB continues to hinder Florida’s citrus industry, it is necessary to create more accurate nutrient application guidelines to improve both HLB-affected tree health and profitability. Although nutrient application methods such as CRF and liquid treatments are likely to lead to higher nutrient uptake compared to 100% soluble granular fertilizers, there is a lack of published literature on nutrient dynamics in HLB-affected grapefruit. Therefore, the goal of this study was to determine if nutrient management methods impact nutrient translocation and to identify where nutrients are translocated within the tree.

## 2. Results

### 2.1. Macronutrient Concentration

Differences in leaf macronutrients were observed between application methods (Figure 1A). The CRF treatment led to 12.1% higher leaf N compared to the LW treatment, higher leaf Ca compared to both the L (27.5%) and LW (25.0%) treatment, and higher S compared to the L (18.25%) and LW (26.43%) treatments. Additionally, 23.8% higher leaf K was observed in the L treatment compared to the control. 

In the twigs, the only difference observed was in Mg levels, where the L treatment led to 37.5% higher Mg compared to the control and 45.2% higher Mg compared to the CRF treatment (Figure 1B).

No differences were observed in macronutrient allocation between treatments in the secondary branches (Figure 1C); however, in the primary branches, K levels were 33.3% higher in the L treatment compared to the control treatment (Figure 1D). In the trunk, the L treatment had 29.0% higher N compared to the LW treatment.

In the structural roots, differences in N and Ca levels were observed between treatments (Figure 2B). The L treatment led to 20% more N compared to the LW treatment. However, the control resulted in 23.1% more Ca compared to the L treatment and 18.8% more Ca compared to the LW treatment. No differences in macronutrients were observed between any of the treatments in the lateral roots (Figure 2C). In the feeder roots, the only difference between macronutrient concentrations was observed in P (Figure 2D). The control led to 19.4% more P compared to the CRF and LW treatments and 34.5% more P compared to the L treatment.

### 2.2. Micronutrient Concentration

In the leaves, B was 50.7% higher in the CRF treatment compared to the L treatment and 42.2% higher compared to the LW treatment (Figure 3A). No differences in micronutrient concentrations were detected between treatments in the twigs (Figure 3B), secondary branches (Figure 3C), primary branches (Figure 3D), trunk (Figure 4A), structural roots (Figure 4B), or the lateral roots (Figure 4C). Like the leaves, B in the feeder roots was 19.8% and 22.2% higher in the CRF treatment compared to the control and LW treatment, respectively (Figure 4D).

### 2.3. Total Tree Nutrient Content and Biomass

No differences in individual tree component (Figure 5) and total N, P, K, Mg, Ca, and S content were observed between any of the treatments (Table 1 and Table 2). Additionally, no differences were observed in total plant N, P, K, Mg, Ca, and S between any of the treatments. Differences in macronutrient levels were observed between different plant components. The greatest amount of total N, P, K, Mg, Ca, and S tended to be found in the leaves, secondary branches, primary branches, and structural roots in all treatments (Table 1 and Table 2). 

Total B, Zn, Mn, and Fe contents per individual tree components were consistent across all treatments except for Mn in the lateral roots (Table 3). The CRF trees had significantly more Mn in the lateral roots compared to the F treatment; however, no differences were observed in total tree micronutrients regardless of treatment (Table 3 and Table 4). 

Differences were observed in total micronutrient contents between plant components. Twigs consistently contained the largest amounts of Zn (61–72% of total plant Zn), Mn (69–79% of total Mn), and Fe (67–71% of total Fe) compared to all other plant components (Table 3 and Table 4). Additionally, the leaves contained between 24–39% of the total tree B in all treatments, making the leaves the greatest pool of B in the entire tree (Table 3). Finally, no differences were observed in total tree dry weight between any treatments (Table 5).

## 3. Discussion

Overall, the application methods showed differences in uptake and translocation, particularly in the leaves and feeder roots. Regardless of the treatment, leaves and feeder roots contained higher concentrations of N and K compared to the other tree components sampled. The leaves also contained higher B concentration compared to any other organ. All other nutrient concentrations were consistent across components.

The CRF treatment had higher N, Ca, S, and B in the leaves compared to the other treatments, and these increases in leaf nutrient concentrations are consistent with the current literature. It is well established that CRF’s outperform conventional granular fertilizers in nutrient uptake and fruit production [27,28,29]. In HLB-affected sweet orange trees, [30] found that CRF formulations resulted in exceptionally high yields. Although no differences in biomass were reported between any of the treatments, the higher macronutrient concentrations in the leaves of the CRF treatment supports the increased effectiveness of CRF’s over conventional fertilizers. Research by [3] did show that a CRF led to higher plant biomass compared to fertigation and water soluble granular treatments; however, the study was conducted on 32-month-old sweet orange trees. The lack of differences observed in total biomass between treatments could be a result of the three-year time frame of this study. This may not have been long enough to detect changes due to fertilizer treatments.

The lower levels of N, Ca, S, and B found in the leaves of the LW treatment compared to the other treatments is important to note. The LW method was chosen as a treatment due to grower reports that supplying nutrients in smaller doses but at shorter intervals led to higher uptake [31]; however, this was not observed. Our results were consistent with those of [32,33] which found that increased fertilizer frequency led to no differences in growth parameters or leaf nutrient concentration of sweet orange trees. The lower uptake of leaf macronutrients in the LW treatment could be due to increased leaching compared to the other treatments. Liquid treatments can lead to a better response compared to granular fertilizers; however, if the grove experiences high rainfall and/or irrigation management is poor, leaching can occur [34]. Additionally, the trees used in the study were severely HLB- affected and had a depleted root system. Logically, a depleted root system will intercept less nutrients and thus the benefits of more frequent fertilization may not be realized.

Higher nutrient contents in the leaves and feeder roots were expected. This was similar to research conducted by [22], which showed that the leaves of 32-month old sweet orange trees had the highest content of total N compared to other tree components. Plants require large amounts of N due to its roles in amino acids, sugars, and proteins [7,35]. In citrus trees, large amounts of N are found in young tender tissues such as leaves [7]. Levels of K are often higher in the leaves due to its role in regulation of stomatal opening and closing [36]. Additionally, citrus fruits utilize large amount of K [3]. Although this study did not sample any fruits due to fruit drop before analysis, the higher contents of K observed in the leaves compared to other plant components could indicate that leaves are a reservoir for K export into fruit. Lastly, the higher levels of B in the leaves compared to other components are likely due to its role in photosynthesis and carbohydrate metabolism [37].

When considering the biomass of each component in relation to nutrient percentage, the leaves, twigs, and secondary branches contained most of the total macronutrients in the trees. Similar findings were reported by [38,39], which showed higher percentages of N in newer organs of citrus trees. The root system, particularly the feeder and lateral roots, contained the lowest macronutrient content. Although the feeder root nutrient samples tend to have higher levels of all nutrients compared to nutrient samples taken from the woody sections of the trees (trunk, branches, lateral and structural roots), the feeder root biomass is much lower compared to the other components. As a result, feeder roots often constitute the lowest amount of nutrients, particularly macronutrients, to the overall nutrient total of the trees.

No other investigators have conducted a destructive study coupled with nutrient analysis of field-grown HLB-affected grapefruit trees. Furthermore, the trees were sub sectioned into 8 different components: five above ground and three below ground components. This level of detail is challenging to obtain due to the inaccessibility of fruit bearing citrus trees that can be removed from the ground and dissected. This provides a greater amount of detailed data on the nutrient uptake and translocation inside the different components and the tree as a whole. Although the aforementioned peculiarities of our study represents strengths, some limitations also need to be acknowledged: (i) the lack of timepoints to compare different seasons and (ii) the method used to remove the trees from the field. Obtaining seasonal timepoints would have required much more resources and available trees to be excavated and dissected over time. This data would have allowed for better inferences regarding the impact of seasonality on nutrient dynamics in HLB-affected citrus trees. On the other hand, the method by which trees were extracted from the field allowed for a greater number of replications to be analyzed but may have resulted in a loss of fine roots. The utilization of a better and more gentle excavation method would have limited fine root loss at the expense of time and resources.

## 4. Materials and Methods

### 4.1. Site Description

A 3-year nutrition trial was conducted at the University of Florida, Institute of Food and Agricultural Sciences (UF/IFAS), Indian River Research and Education Center (IRREC) located in Fort Pierce, Florida, USA. Tree material consisted of 6-year-old field grown ‘Ruby Red’ grapefruit trees (*Citrus* × *paradisi*) grafted on ‘sour orange’ (*Citrus* × *aurantium*) rootstock (Figure 6). All trees examined were HLB-affected, confirmed by both Ct values and visual analysis.

The trees were grown on 1-m-high raised beds for drainage purposes and irrigated with 39.7 L per hour microjet sprinklers (Maxijet, Dundee, FL, USA). The grove soils were sandy Alfisols classified as loamy, siliceous, active, hyperthermic Arenic Glossaqualfs with less than 1% organic matter. Soil pH was 5.8 and cation exchange capacity (C.E.C.) was 3.5 cmol kg^−1^ [40].

Treatments consisted of four different fertilizer application methods (Control, controlled release fertilizer (CRF), liquid (L) and liquid weekly (LW). The control treatment was an industry standard 100% soluble dry granular fertilizer applied three times throughout the growing season. The CRF treatment was a granular fertilizer with 50% solubility and contained polymer coated nutrients. Like the control treatment, the CRF treatment was applied three times throughout the growing season. The L and LW were both liquid treatments applied at a rate of 11.36 L per tree using a ~1100-L single axle admire mobile spray tank (Chemical Containers, Inc., Lake Wales, FL, USA). The L treatment was applied bi-weekly throughout the growing season while the LW treatment was applied weekly. The same yearly amount of nutrients were applied to all treatments regardless of application method. These amounts were calculated using the current UF/IFAS recommendations [41].

### 4.2. Tree Excavation and Dissection

Trees were excavated in September 2021 using a John Deere 7030 tractor and root rake implement. The base of the tree trunk was securely grasped by the rake and slowly lifted from the soil. During the lifting process, the trees were gently shaken to assist in removal of sand from the root mass. Once removed from the soil, the entire tree was moved to a covered location for dissection.

Each tree was divided into eight different components: leaf, twig, secondary branch, primary branch, trunk, structural root, lateral root, and feeder root (Figure 7). All leaves, including both immature and fully expanded leaves, were collected by hand. Twigs were defined as the woody portions connected to the leaves. Secondary branches were defined as those directly connected to the twigs but not directly connected to the main trunk. The primary branches were those that connected directly to the main trunk and supported the secondary branches. The main trunk was defined as the central woody structure connecting the root mass to the branches.

Roots were divided into structural, lateral, and feeder roots. The structural roots were directly connected to the trunk and ranged from 10 mm to 20 mm. Lateral roots connected the primary roots to the feeder roots and ranged from 2 mm to 10 mm. Feeder roots were defined as the non-woody roots less than 2 mm in diameter protruding from the lateral roots.

The different components were removed from one another using Felco 2 pruners (Les Geneveys-sur-Coffrane, Switzerland), Fiskars 28″ loppers (Helsinki, Finland), and a Ryobi 40V HP Brushless 14in. battery operated chainsaw (Fuchu, Hiroshima, Japan). Once the components were removed from the tree and separated from each other, the total weight of each component, as well as nutrient analysis, was conducted.

### 4.3. Tree Biomass and Mineral Analysis

The entire fresh weight of each component was collected and weighed using a digital field scale (Ohaus Corporation, Parsippany, NJ, USA). Subsamples were then collected from each component and weighed. The subsamples were then dried at 60 °C for 3 to 15 days depending on the component size. Once dried, the subsample dry weight was collected using an analytical scale (Sartorius AG, Göttingen, Germany). Dried subsamples were then ground to pass through 1.0 mm mesh screen and 5 mL of nitric acid (HNO_3_) was added. Samples were then heated to 95 °C for 90 min and 4 mL of 30% Hydrogen Peroxide (H_2_O_2_) was added. After 20 min of cooling, 50 mL of deionized water was added to each sample. Analysis of N, P, K, Mg, Ca, S, B, Zn, Mn, and Fe concentration was conducted using inductively coupled argon plasma emission (ICP-MS) spectrophotometer (Spectro Ciros CCD, Fitzburg, MA, USA) [42].

### 4.4. Experimental Design and Statistical Analysis

The experiment was organized into a completely randomized design with split plot arrangement. Each treatment was replicated 4 times and each replicant consisted of 10 trees. One tree from each replication was randomly selected to be excavated and dissected. A one-way Analysis of Variance (ANOVA) was used to determine significant differences between fertilizer treatments. When differences were detected (*p* < 0.05), a Tukey’s honestly significant difference (HSD) test was conducted. A Kruskal–Wallis test was used in combination with a Tukey’s HSD test to identify and test significant differences in total nutrients in individual tree components. All statistical analyses were conducted using the software R with ‘agricolae’ package [43]. Figures were generated using the software Minitab 17 (Minitab, LCC, State College, PA, USA).

## 5. Conclusions

Although nutrient use has been extensively studied in citrus, the uptake and distribution of nutrients in field grown HLB-affected grapefruit trees is less understood. This was the first study to conduct destructive nutrient analysis on field grown HLB-affected grapefruit trees. The research above shows where in the trees nutrients are stored and how much of each given nutrient are present in the entire tree. Additionally, this study showed that fertilization methods do impact nutrient contents in different components of HLB-affected trees. With these results in mind, further research should be conducted on how foliar nutrient application methods impact nutrient allocation within HLB-affected trees. Future studies should include internal movements of nutrients and tree destructive analysis at multiple time points throughout the year to account for seasonal variations. These continued evaluations of nutrient uptake and allocation could improve the efficiency of nutrient management programs in the age of HLB.

## Figures and Tables

**Figure 1 plants-11-03226-f001:**
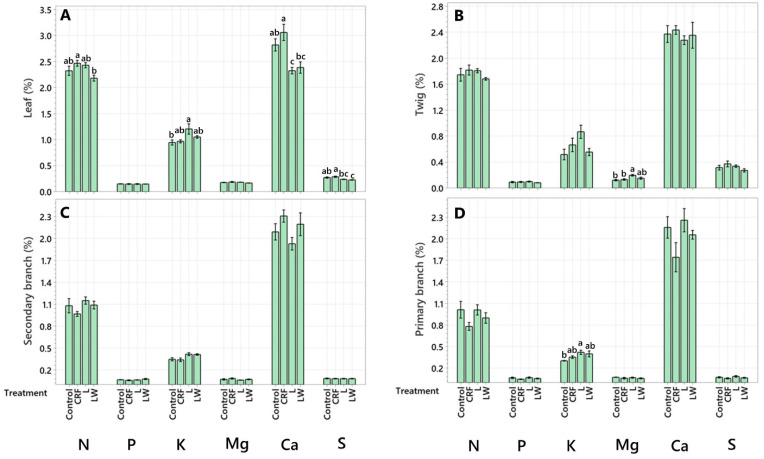
Macronutrient concentration (%) in leaf (**A**), twig (**B**), secondary branch (**C**), and primary branch (**D**). Six-year-old Huanglongbing-affected ‘Ruby Red’ grapefruit trees grafted on sour orange rootstock were used. Treatments consisted of two liquid (L and LW) and two granular fertilizers (Control and controlled-release fertilizer (CRF)). Treatments were applied three times a year (control and CRF), biweekly (L), or weekly (LW), for three years. A one-way variance (ANOVA) with a Tukey’s honestly significant difference (HSD) test was used to determine significant differences between means. Lowercase letters (a, b, c) indicate statistically significant differences (*p* ≤ 0.05). Data represents means (*n* = 4) ± standard error.

**Figure 2 plants-11-03226-f002:**
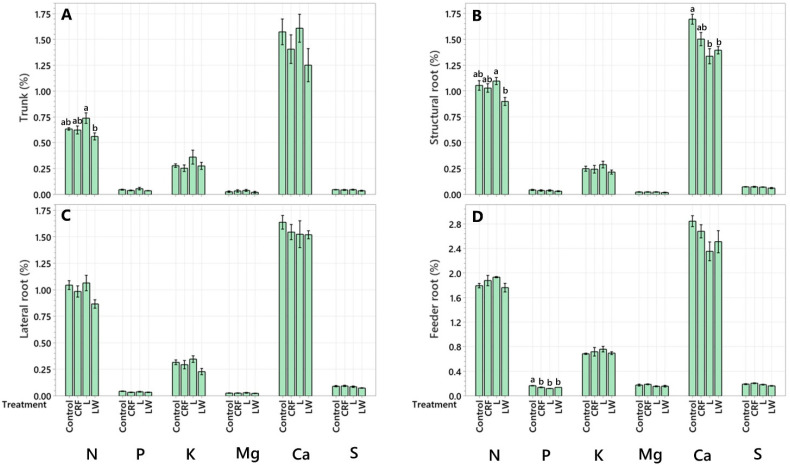
Macronutrient concentration (%) in trunk (**A**), structural root (**B**), lateral root (**C**), and feeder root (**D**). Six-year-old Huanglongbing-affected ‘Ruby Red’ grapefruit trees grafted on sour orange rootstock were used. Treatments consisted of two liquid (L and LW) and two granular fertilizers (Control and controlled-release fertilizer (CRF)). Treatments were applied three times a year (control and CRF), biweekly (L), or weekly (LW), for three years. A one-way analysis of variance (ANOVA) with a Tukey’s honestly significant difference (HSD) test was used to determine significant differences between means. Lowercase letters (a, b) indicate statistically significant differences (*p* ≤ 0.05). Data represents means (*n* = 4) ± standard error.

**Figure 3 plants-11-03226-f003:**
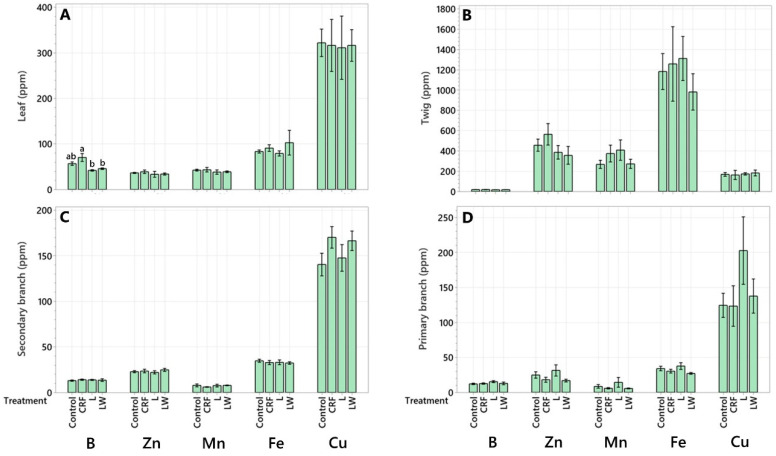
Micronutrient concentration (ppm) in leaf (**A**), twig (**B**), secondary branch (**C**), and primary branch (**D**). Six-year-old Huanglongbing-affected ‘Ruby Red’ grapefruit trees grafted on sour orange rootstock were used. Treatments consisted of two liquid (L and LW) and two granular fertilizers (Control and controlled-release fertilizer (CRF)). Treatments were applied three times a year (control and CRF), biweekly (L), or weekly (LW), for three years. A one-way variance (ANOVA) with a Tukey’s honestly significant difference (HSD) test was used to determine significant differences between means. Lowercase letters (a, b) indicate statistically significant differences (*p* ≤ 0.05). Data represents means (*n* = 4) ± standard error.

**Figure 4 plants-11-03226-f004:**
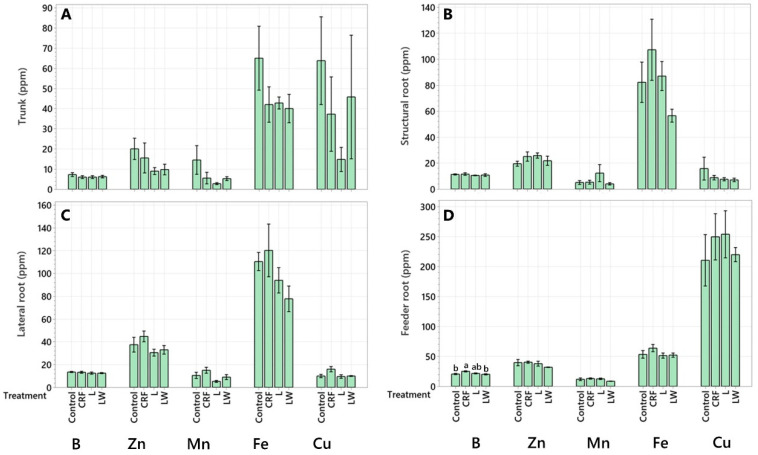
Micronutrient concentration (ppm) in trunk (**A**), structural root (**B**), lateral root (**C**), and feeder root (**D**). Six-year-old Huanglongbing-affected ‘Ruby Red’ grapefruit trees grafted on sour orange rootstock were used. Treatments consisted of two liquid (L and LW) and two granular fertilizers (Control and controlled-release fertilizer (CRF)). Treatments were applied three times a year (control and CRF), biweekly (L), or weekly (LW), for three years. A one-way variance (ANOVA) with a Tukey’s honestly significant difference (HSD) test was used to determine significant differences between means. Lowercase letters (a, b) indicate statistically significant differences (*p* ≤ 0.05). Data represents means (*n* = 4) ± standard error.

**Figure 5 plants-11-03226-f005:**
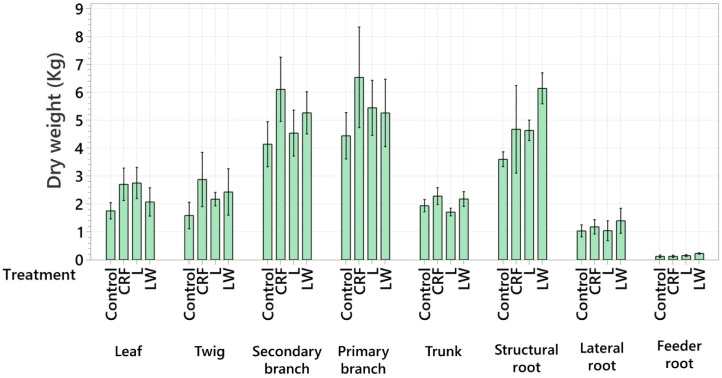
Dry weight (kg) of each tree component. Six-year-old Huanglongbing-affected ‘Ruby Red’ grapefruit trees grafted on sour orange rootstock were used. Treatments consisted of two liquid (L and LW) and two granular fertilizers (Control and controlled-release fertilizer (CRF)). Treatments were applied three times a year (control and CRF), biweekly (L), or weekly (LW), for three years. Data represents means (*n* = 4) ± standard error.

**Figure 6 plants-11-03226-f006:**
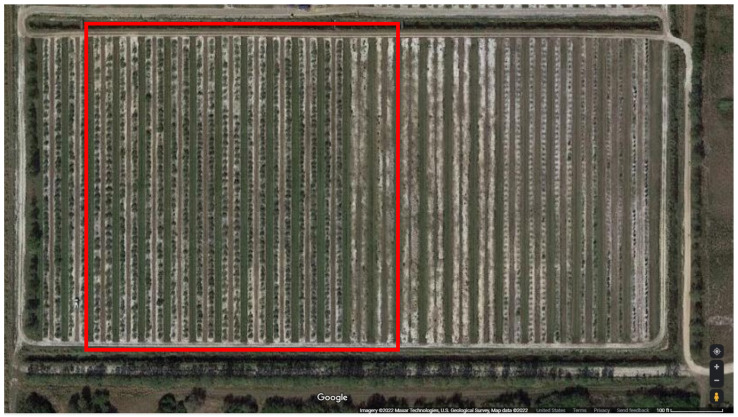
Satellite image of the area in which the study was conducted. The experimental grove from which the grapefruit trees were excavated is highlighted in red. The grapefruit grove is located at the University of Florida, Institute of Food and Agricultural Sciences (UF/IFAS), Indian River Research and Education Center (IRREC) located in Fort Pierce, Florida, USA. Image was acquired from Google Maps on 16 November 2022.

**Figure 7 plants-11-03226-f007:**
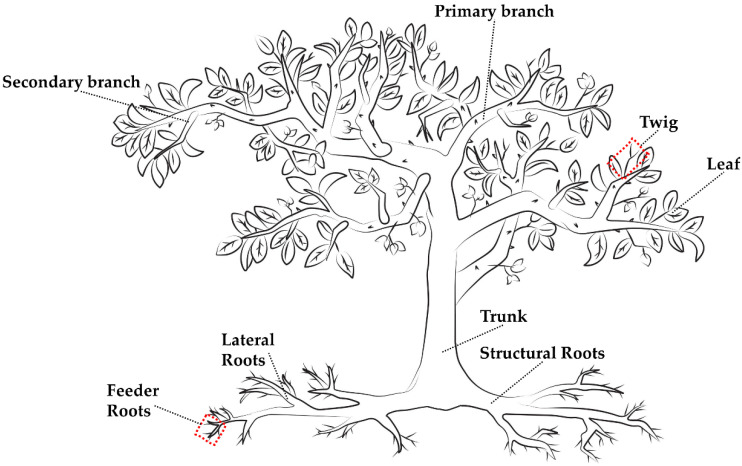
Schematic representation of a six-year-old grapefruit tree divided into eight different components: leaf, twig, secondary branch, primary branch, trunk, structural root, lateral root, and feeder root.

**Table 1 plants-11-03226-t001:** Nitrogen (N), phosphorus (P), and potassium (K) contents per individual tree components and tree total. Six-year-old Huanglongbing -affected ‘Ruby Red’ grapefruit trees grafted on sour orange rootstock were used. Treatments consisted of two liquid (L and LW) and two granular fertilizers (Control and controlled-release fertilizer (CRF)). A Kruskal–Wallis test was used in combination with a Tukey’s honestly significant difference (HSD) test to identify and test significant differences in total nutrients in individual tree components (differences signified by letters). Data represents means (*n* = 4) ± standard error.

	**N**			
		g tree^−1^		
Tree component	Control	CRF	L	LW
Leaf	40.64 ± 7.14 abc	67.51 ± 14.9 a	66.42 ± 13.31 a	45.35 ± 10.88 abc
Twig	26.81 ± 7.11 abcd	50.75 ± 15.27 abc	39.4 ± 4.89 ab	40.89 ± 13.86 abc
Secondary Branch	46.67 ± 13.03 ab	59.59 ± 12.84 ab	51.57 ± 8.57 a	56.72 ± 7.17 a
Primary Branch	47.43 ± 12.17 a	48.55 ± 9.36 abc	54.07 ± 8.53 a	48.38 ± 14.41 ab
Trunk	12.34 ± 1.51 bcd	14.34 ± 2.16 bc	12.67 ± 1.33 bc	12.40 ± 1.99 bc
Structural Root	38.31 ± 4.22 abc	46.38 ± 15.56 abc	50.68 ± 3.46 a	55.69 ± 6.78 a
Lateral Root	10.68 ± 2.22 cd	11.30 ± 2.22 bc	10.50 ± 2.87 bc	12.40 ± 4.37 bc
Feeder Root	2.15 ± 0.84 d	2.19 ± 0.76 c	2.88 ± 0.65 c	3.9 ± 0.24 c
Total	225.07 ± 1.81	300.64 ± 5.55	288.22 ± 5.17	275.77 ± 4.60
	**P**			
		g tree^−1^		
Tree component	Control	CRF	L	LW
Leaf	2.6 ± 0.40 ab	4.00 ± 0.87 a	4.11 ± 0.97 a	3.0 ± 0.76 a
Twig	1.34 ± 0.31 abc	2.78 ± 0.91 ab	2.2 ± 0.41 ab	1.98 ± 0.65 abc
Secondary Branch	2.83 ± 0.68 ab	3.87 1.18 a	2.95 ± 0.58 ab	3.91 ± 0.31 a
Primary Branch	3.14 ± 0.77 a	2.96 ± 0.62 ab	4.00 ± 1.09 a	2.94 ± 0.72 ab
Trunk	0.93 ± 0.16 bc	0.91 ± 0.15 ab	0.98± 0.24 b	0.81± 0.11 bc
Structural Root	1.64 ± 0.28 abc	1.90 ± 0.82 ab	1.88 ± 0.44 ab	2.02 ± 0.40 abc
Lateral Root	0.46 ± 0.09 c	0.42 ± 0.10 b	0.45 ± 0.20 b	0.47 ± 0.13 c
Feeder Root	0.2 ± 0.08 c	0.16 ± 0.05 b	0.18 ± 0.04 b	0.31 ± 0.03 c
Total	13.19 ± 0.23	17.03 ± 1.50	16.78 ± 1.45	15.53 ± 3.38
	**K**			
		g tree^−1^		
Tree component	Control	CRF		L
Leaf	16.33 ± 2.55 a	26.19 ± 5.58 a		33.91 ± 8.82 a
Twig	7.59 ± 2.23 bcd	19.17 ± 6.31 abc		19.4 ± 4.48 abc
Secondary Branch	13.88 ± 1.90 ab	21.04 ± 4.92 ab		18.96 ± 3.55 ab
Primary Branch	13.64 ± 2.67 abc	22.52 ± 5.22 ab		23.63 ± 5.69 abc
Trunk	5.52 ± 0.95 cd	5.83 ± 1.06 bc		6.18 ± 1.16 bc
Structural Root	9.05 ± 1.16 abcd	10.04 ± 3.64 abc		13.36 ± 1.63 bc
Lateral Root	3.29 ± 0.66 d	3.33 ± 0.70 bc		3.59 ± 1.22 bc
Feeder Root	0.83 ± 0.32 d	0.78 ± 0.25 c		1.17 ± 0.29 c
Total	70.17 ± 0.58	108.96 ± 3.93		120.25 ± 4.01

**Table 2 plants-11-03226-t002:** Magnesium (Mg), calcium (Ca), and potassium (S) contents per individual components and tree total. Six-year-old Huanglongbing-affected ‘Ruby Red’ grapefruit trees grafted on sour orange rootstock were used. Treatments consisted of two liquid (L and LW) and two granular fertilizers (Control and controlled-release fertilizer (CRF)). A Kruskal–Wallis test was used in combination with a Tukey’s honestly significant difference (HSD) test to identify and test significant differences in total nutrients in individual tree components (differences signified by letters). Data represents means (*n* = 4) ± standard error.

	**Mg**			
		g tree^−1^		
Tree component	Control	CRF	L	LW
Leaf	3.09 ± 0.50 a	5.17 ± 1.22 a	4.98 ± 0.97 a	3.43 ± 0.78 a
Twig	1.89 ± 0.59 abc	3.5 ± 0.91 abc	4.33 ± 0.68 a	3.59 ± 1.03 a
Secondary Branch	2.97 ± 0.71 ab	5.19 ± 1.07 a	2.79 ± 0.44 a	3.93 ± 0.87 a
Primary Branch	3.33 ± 0.60 a	3.66 ± 0.51 ab	3.65 ± 0.47 a	3.02 ± 0.67 ab
Trunk	0.57 ± 0.22 c	0.82 ± 0.30 bc	0.67 ± 0.16 bc	0.49 ± 0.19 b
Primary Root	0.90 ± 0.12 bc	1.10 ± 0.41 bc	1.19 ± 0.22 bc	1.24 ± 0.32 ab
Secondary Root	0.27 ± 0.04 c	0.31 ± 0.06 c	0.31 ± 0.11 c	0.31 ± 0.07 b
Feeder Root	0.22 ± 0.09 c	0.22 ± 0.07 c	0.23 ± 0.05 c	0.35 ± 0.03 b
Total	13.268 ± 0.33	20.01 ± 1.96	18.17 ± 1.90	16.40 ± 1.67
	**Ca**			
		g tree^−1^		
Tree component	Control	CRF	L	LW
Leaf	49.71 ± 9.34 abc	83.65 ± 18.45 abc	63.39 ± 12.60 bc	47.77 ± 9.53 bcd
Twig	36.42 ± 9.76 bc	71.09 ± 24.22 abcd	49.9 ± 7.06 bcd	54.23 ± 15.39 abcd
Secondary Branch	85.05 ± 14.59 a	138.67 ± 23.22 a	87.72 ± 17.63 ab	118.75 ± 25.01 a
Primary Branch	97.40 ± 19.79 a	103.83 ± 13.01 ab	122.57 ± 22.45 a	106.77 ± 23.41 ab
Trunk	30.46 ± 3.73 bc	32.32 ± 5.41 bcd	27.63 ± 3.29 cd	27.69 ± 5.44 cd
Structural Root	60.93 ± 4.34 ab	72.91 ± 24.80 abcd	62.56 ± 7.43 bc	85.98 ± 9.17 abc
Lateral Root	16.99 ± 3.68 bc	18.38 ± 4.14 cd	15.64 ± 5.04 cd	21.44 ± 7.05 cd
Feeder Root	3.54 ± 1.46 c	3.15 ± 1.06 d	3.34 ± 0.65 d	5.59 ± 0.61 d
Total	380.53 ± 1.24	524.05 ± 8.08	432.79 ± 7.60	468.24 ± 7.57
	**S**			
		g tree^−1^		
Tree component	Control	CRF	L	LW
Leaf	4.78 ± 0.87 a	7.96 ± 1.90 ab	6.59 ± 1.39 ab	4.64 ± 1.01 ab
Twig	4.92 ± 1.52 a	9.69 ± 2.31 a	7.42 ± 1.19 a	6.19 ± 1.72 a
Secondary Branch	3.63 ± 0.89 ab	5.05 ± 1.09 abc	3.67 ± 0.57 bc	4.39 ± 0.76 abc
Primary Branch	3.49 ± 0.87 ab	3.54 ± 0.44 bc	4.73 ± 1.02 ab	3.55 ± 1.05 abc
Trunk	0.93 ± 0.13 b	1.03 ± 0.17 c	0.81 ± 0.09 c	0.82 ± 0.15 bc
Structural Root	2.71 ± 0.27 ab	3.43 ± 1.19 bc	3.37 ± 0.34 bc	3.87 ± 0.63 abc
Lateral Root	0.92 ± 0.16 b	1.07 ± 0.20 c	0.88 ± 0.27 c	1.03 ± 0.30 abc
Feeder Root	0.22 ± 0.07 b	0.24 ± 0.08 c	0.27 ± 0.06 c	0.36 ± 0.04 c
Total	21.63 ± 0.53	32.04 ± 2.50	27.78 ± 2.12	24.89 ± 1.63

**Table 3 plants-11-03226-t003:** Boron (B), zinc (Zn), and manganese (Mn) contents per individual components and tree total. Six-year-old Huanglongbing-affected ‘Ruby Red’ grapefruit trees grafted on sour orange rootstock were used. Treatments consisted of two liquid (L and LW) and two granular fertilizers (Control and controlled-release fertilizer (CRF)). A Kruskal–Wallis test was used in combination with a Tukey’s honestly significant difference (HSD)test to identify and test significant differences in total nutrients in individual tree components (differences signified by letters). A one-way analysis of variance (ANOVA) with a Tukey HSD test was used to determine significant differences between the means of between different fertilizer application methods (differences signified by *). Data represents means (*n* = 4) ± standard error.

	**B**			
		mg kg^−1^		
Tree component	Control	CRF	L	LW
Leaf	99.77 ± 20.85 a	199.19 ± 58.39 a	115.65 ± 24.16 a	92.58 ± 20.11 a
Twig	29.16 ± 9.14 bc	52.38 ± 17.25 b	35.68 ± 3.57 bcd	41.02 ± 12.78 abc
Secondary Branch	52.70 ± 8.74 b	85.73 ± 18.14 b	61.16 ± 9.01 bc	74.04 ± 17.64 ab
Primary Branch	55.58 ± 13.56 ab	77.05 ± 14.13 b	81.80 ± 14.40 ab	70.92 ± 26.73 abc
Trunk	14.00 ± 2.14 bc	13.77 ± 2.40 b	10.28 ± 1.43 d	13.69 ± 2.10 bc
Structural Root	40.42 ± 3.32 bc	56.37 ± 20.04 b	48.99 ± 5.16 bcd	66.23 ± 9.86 abc
Lateral Root	13.67 ± 2.58 bc	15.54 ± 3.43 b	12.51 ± 3.65 cd	17.44 ± 5.42 bc
Feeder Root	2.35 ± 0.82 c	3.09 ± 1.15 b	3.22 ± 0.73 d	4.43 ± 0.35 c
Total	307.70 ± 1.77	503.17 ± 9.18	369.32 ± 6.85	380.40 ± 5.79
	**Zn**			
		mg kg^−1^		
Tree component	Control	CRF	L	LW
Leaf	62.45 ± 8.47 b	109.01 ± 29.73 b	93.43 ± 23.72 b	71.35 ± 20.47 b
Twig	756.91 ± 254.33 a	1467.75 ± 403.15 a	815.33 ± 128.33 a	870.69 ± 342.09 a
Secondary Branch	92.43 ± 15.89 b	142.7 6± 30 31 b	96.53 ± 13.51 b	127.33 12.56 b
Primary Branch	113.91 ± 27.63 b	102.58 ± 11.14 b	169.62 ± 54.63 b	86.82 ± 24.17 b
Trunk	37.67 ± 9.94 b	36.44 ± 17.89 b	15.70 ± 3.66 b	21.72 ± 6.13 b
Structural Root	70.97 ± 9.83 b	127.51 ± 45.64 b	119.87 ± 14.28 b	131.17 ± 21.74 b
Lateral Root	34.87 ±c4.05 b	49.10 ± 7.81 b	28.78 ± 6.13 b	43.17 ± 10.13 b
Feeder Root	4.46 ± 1.39 b	5.05 ± 1.93 b	5.39 ± 1.13 b	7.08 ± 0.65 b
Total	1173.31 ± 7.36	2040.23 ± 31.68	1344.69 ± 21.45	1359.36 ± 22.08
	**Mn**			
		mg kg^−1^		
Tree component	Control	CRF	L	LW
Leaf	73.07 ± 11.10 b	121.10 ± 34.10 b	104.64 ± 22.53 b	80.04 ± 19.03 b
Twig	453.87 ± 157.22 a	931.26 ± 265.62 a	843.84 ± 157.15 a	686.03 ± 246.98 a
Secondary Branch	32.95 ± 11.28 b	35.23 ± 4.32 b	32.16 ± 7.24 b	39.76 ± 3.60 b
Primary Branch	38.26 ± 10.90 b	32.56 ± 1.24 b	79.98 ± 44.31 b	29.90 ± 9.83 b
Trunk	27.62 ± 13.46 b	12.91 ± 6.97 b	4.82 ± 1.13 b	11.43 ± 2.36 b
Structural Root	18.95 ± 5.85 b	23.25 ± 11.78 b	61.36 ± 36.51 b	23.47 ± 4.25 b
Lateral Root *	9.60 ± 2.30 b	16.26 ± 3.50 b	4.67 ± 0.69 b	9.98 ± 1.15 b
Feeder Root	1.26 ± 0.33 b	1.67 ± 0.66 b	1.75 ± 0.32 b	1.89 ± 0.17 b
Total	655.63 ± 6.34	1174.27 ± 29.55	1133.25 ± 26.47	882.53 ± 24.53

**Table 4 plants-11-03226-t004:** Iron (Fe) and copper (Cu) contents per individual components and tree total. Six-year-old Huanglongbing-affected ‘Ruby Red’ grapefruit trees grafted on sour orange rootstock were used. Treatments consisted of two liquid (L and LW) and two granular fertilizers (Control and controlled-release fertilizer (CRF)). A Kruskal–Wallis test was used in combination with a Tukey’s honestly significant difference (HSD) test to identify and test significant differences in total nutrients in individual tree components (differences signified by letters). Data represents means (*n* = 4) ± standard error.

	**Fe**			
		mg kg^−1^		
Tree component	Control	CRF	L	LW
Leaf	144.04 ± 21.07 b	251.31 ± 59.80 b	215.68 ± 40.79 b	218.14 ± 73.26 b
Twig	2060 ± 863.63 a	3305.2 ± 1182.37 a	2827.4 ± 507.11 a	2380.39 ± 818.77 a
Secondary Branch	140.61 ± 22.13 b	197.23 ± 33.46 b	149.03 ± 28.58 b	170.09 ± 25.40 b
Primary Branch	151.01 ± 28.07 b	204.10 ± 67.91 b	197.80 ± 35.09 b	137.59 ± 25.97 b
Trunk	122.70 ± 29.65 b	98.34 ± 24.10 b	74.21 ± 10.74 b	87.74 ±17.99 b
Structural Root	304.55 ± 77.35 b	473.65 ± 202.92 b	403.70 ± 57.21 b	342.15 ± 31.18 b
Lateral Root	115.10 ± 24.12 b	123.89 ± 14.00 b	88.32 ± 18.88 b	104.99 ± 29.36 b
Feeder Root	6.22 ± 2.07 b	7.34 ± 2.68 b	7.31 ± 1.53 b	11.49 ± 0.86 b
Total	3044.27 ± 16.42	4661.09 ± 47.00	3963.51 ± 43.75	3452.63 ± 39.62
	**Cu**			
		mg kg^−1^		
Tree component	Control	CRF	L	LW
Leaf	551.08 ± 77.48 a	908.36 ± 310.29 a	880.74 ± 249.09 a	660.71 ± 194.10 ab
Twig	260.7 ± 69.64 ab	425.37 ± 149.26 abc	377.82 ± 52.99 ab	377.4 ± 94.32 ab
Secondary Branch	574.28 ± 109.89 a	1000.56 ± 127.06 a	653.89 ± 114.90 ab	859.96 ± 102.49 a
Primary Branch	570.21 ± 131.41 a	663.72 ± 81.37 ab	1113.84 ± 346.38 a	790.24 ± 325.53 a
Trunk	123.32 ± 41.76 b	86.529 ± 44.81 bc	25.97 ± 11.06 b	98.32 ± 62.54 b
Structural Root	58.39 ± 32.95 b	42.88 ± 16.66 bc	35.77 ± 8.15 b	40.88 ± 3.62 b
Lateral Root	9.49 ± 1.34 b	17.98 ± 4.28 c	9.83 ± 3.34 b	13.64 ± 3.90 b
Feeder Root	22.64 ± 6.00 b	33.97 ± 14.97 c	35.89 ± 8.29 b	49.85 ± 7.69 b
Total	2170.15 ± 198.15	3179.4 ± 1649.66	3133.80 ± 1735.76	2891.04 ± 1520.87

**Table 5 plants-11-03226-t005:** Total mean dry weight (DW) per treatment. Six-year-old HLB-affected ‘Ruby Red’ grapefruit trees grafted on sour orange rootstock were used. Treatments consisted of two liquid (L and LW) and two granular fertilizers (Control and controlled-release fertilizer (CRF)). Data represents means ± standard error.

Treatment	Total (DW)
	kg
Control	18.61 ± 1.81
CRF	26.47 ± 4.88
L	18.45 ± 2.74
LW	20.98 ± 3.99

## Data Availability

Raw data will be available by requesting them via email to l.rossi@ufl.edu.

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
