# Peer review of "Root-Shoot Nutrient Dynamics of Huanglongbing-Affected Grapefruit Trees"

_plants, 2022, doi:10.3390/plants11233226_

Round 1

Reviewer 1 Report

Lines 341-343 the authors need to descripe the method used to extract the nutrients from the plant ground tissue . (eg the acid solution used)

Although the results presented here are interesting, the authors didn’t investigate the internal movement of nutrients. In addition, the results might be more interesting if the authors analyze the plant tissues, during  the three year period, by sampling at different time points. 

Author Response

Lines 341-343 the authors need to descripe the method used to extract the nutrients from the plant ground tissue . (eg the acid solution used)

Thank you for your suggestion. Details regarding the method of extraction for ground plant tissue was added to line 330-333.

Although the results presented here are interesting, the authors didn’t investigate the internal movement of nutrients. In addition, the results might be more interesting if the authors analyze the plant tissues, during the three year period, by sampling at different time points. 

Thank you for your suggestions. Although we did not analyze internal movement of nutrients and nutrient contents in plant tissues at multiple time points, we suggested this for future studies (line 364-365).

Reviewer 2 Report

The authors have studied the Root-Shoot Nutrient Dynamics of Huanglongbing-Affected Grapefruit Trees. Please see my suggestions below: 

L40. The final of the paragraph needs references; in this regard, I suggest checking and referring to the following two papers:  https://doi.org/10.1007/s11356-019-04214-1

 https://doi.org/10.1007/s11356-021-14127-7

Extensive Results part, many results presented. However, all 4 figures in the Figure 1 are blurred. Please provide better (best) quality figures. Similar for all the other figures in the manuscript.

 Please also replace “Letters” in 109 with “Lowercase letters (a, b).

Under each table, please explain all abbreviations used in them.

Paragraph 224-227 must be removed as it explains in duplicate the aim of the study, already described in L78-80. I suggest choosing the best/complete description of the aim of the study, keeping it as correctly was inserted at the final of Introduction section.

After L281, please describe in a paragraph the strength and limitations of your research.

Subsection 4.1. A satellite photo of the site would be relevant.

L332. Please remove the line (endash) before “Shematic…”

Subsection 4.4. Please mention all the computer programs and their variants used for the statistical analysis.

L353. The link mentioned at the final of L353 must be mentioned/inserted in the References section and inserted here only as a number [41]. In References, please also insert the date of accessing this link.

Author Response

L40. The final of the paragraph needs references; in this regard, I suggest checking and referring to the following two papers:  https://doi.org/10.1007/s11356-019-04214-1 https://doi.org/10.1007/s11356-021-14127-7

These citations have been added to the manuscript.

Extensive Results part, many results presented. However, all 4 figures in the Figure 1 are blurred. Please provide better (best) quality figures. Similar for all the other figures in the manuscript.

Figures have been updated to a higher resolution.

Please also replace “Letters” in 109 with “Lowercase letters (a, b).

“Lowercase letters (a, b) was added to the figure descriptions.

Under each table, please explain all abbreviations used in them.

All abbreviations under figures and tables were explained.

Paragraph 224-227 must be removed as it explains in duplicate the aim of the study, already described in L78-80. I suggest choosing the best/complete description of the aim of the study, keeping it as correctly was inserted at the final of Introduction section.

This section was removed from the manuscript.

After L281, please describe in a paragraph the strength and limitations of your research.

Strengths and limitations have been added to the paper.

Subsection 4.1. A satellite photo of the site would be relevant.

A satellite photo has been added.

L332. Please remove the line (endash) before “Shematic…”

The line was removed.

Subsection 4.4. Please mention all the computer programs and their variants used for the statistical analysis.

R was cited and the statistical package used was included.

L353. The link mentioned at the final of L353 must be mentioned/inserted in the References section and inserted here only as a number [41]. In References, please also insert the date of accessing this link.

The link has been removed and the citation has been added in its place.

Round 2

Reviewer 2 Report

The authors have responded to my suggestions.